# Vaccination against H5 HP avian influenza virus leads to persistent immune response in wild king penguins

Mathilde Lejeune[1], Jérémy Tornos [1], Tristan Bralet[1,2], Camille De Pasquale [1], Elsa Marçon [3], Pascale Massin [4], Béatrice Grasland[4], Antoine Stier[3,5] & Thierry Boulinier [1] ✉

Since 2021, the panzootic nature of high pathogenicity avian influenza (HPAI) represents an increasing threat to wild vertebrate populations. In this context, recent vaccines developed for poultry could provide tools for the conservation of wild endangered birds populations. The king penguin (*Aptenodytes patagonicus*), a long-lived seabird breeding in dense colonies with an extended chick-rearing period, has been identified as a possible surrogate species for a vaccination trial in a sub-Antarctic natural setting. Here we investigate the immune response of king penguin chicks to a self-amplifying mRNA vaccine against a H5 HPAI clade 2.3.4.4b protein. The cohort entails thirty vaccinated chicks (primo- and boost-injections), and 20 unvaccinated controls. Along 250 days of monitoring, the vaccinated chicks show a high and persistent immune response, granting a strong sero-neutralisation capacity against the virus, up to fledging. No adverse effects are observed. Screening for antibodies against unspecific avian influenza viruses suggests that no natural infection has occurred over the entire trial. The emergence of HPAI in the Southern Indian Ocean in October 2024 highlights the timeliness of such experimental tests. Our results thus show the vaccine could provide a potentially powerful tool for mitigation of avian flu outbreaks in the wild.

High pathogenicity avian influenza (HPAI) H5Nx has become a critical threat to biodiversity, affecting hundreds of species and hundreds of thousands of individuals of wild species[1,2]. Since 2021, H5Nx clade 2.3.4.4b HPAI viruses have been responsible for a panzootic that has spread to all continents except Australia, representing an increasing threat to wild species, domestic animals and humans[3]. HPAI H5Nx viruses have emerged from farmed poultry, spreading globally. They are associated with high mortality rates, leading to massive die-offs in densely breeding species, and could potentially threaten small populations with extinction[4–6]. Among newly affected populations, colonial long-lived species such as seabirds[4,7–9], condors[10], vultures[11] and pinnipeds[5], are highly sensitive to adult mortality[12]. This is notably relevant for remote and threatened populations, such as those from Antarctic and sub-Antarctic oceanic regions, where the HPAI H5N1 virus has been first detected in 2023[13,14]. In such areas, the spread of the virus could induce an additional threat to marine vertebrate populations, already under pressure from invasive species, fishing activities or climate change[15,16]. In particular, penguin populations appear to be of prime concern[17].

For threatened populations, vaccination has been identified as a potentially powerful mitigation and protective measure against HPAI

[1]Centre d'Ecologie Fonctionnelle et Evolutive (CEFE), CNRS, Université Montpellier, EPHE, IRD, Montpellier, France. [2]ANSES, Animal Health Laboratory, Bacterial Zoonoses Unit, Maisons-Alfort, France. [3]Université de Strasbourg, CNRS, IPHC UMR 7178, Strasbourg, France. [4]ANSES, Laboratoire de Ploufragan-Plouzané-Niort, Laboratoire National de Référence pour l'influenza aviaire, Unité de virologie, Immunologie, Parasitologie, Aviaires et Cunicoles, Ploufragan, France. [5]Department of Biology, University of Turku, Turku, Finland. ✉e-mail: thierry.boulinier@cefe.cnrs.fr

in emergency situations[18]. However, a critical challenge is the limited availability of scientific data on the suitability of vaccines for wild species in natural settings[10]. Studies have investigated HPAI vaccination in wild species held in captive conditions, notably in zoos following European Commission directives resulting from first HPAI epidemics in the early 2000s. The use of heterologous inactivated vaccines for captive birds appeared to induce a strong immune response, although temporary and heterogeneous among species[19–24]. Serological studies notably suggested that captive Sphenisciformes (penguins) did not show a strong immune response following vaccination using inactivated vaccines, while flamingos developed an antibody response lasting up to seven years[25,26]. Most of the vaccines used in this context are nevertheless not homologous with the currently circulating viruses from the H5Nx 2.3.4.4b clade. Two decades later, following the arrival of HPAI in South Africa in 2022, an investigation was conducted in adult captive individuals of the critically endangered African penguins (*Spheniscus demersus*) using two types of vaccines (an inactivated H5N8 and a H5 avian influenza virus-like particle vaccine)[27]. This latter study highlighted a potential antibody persistence over several months in this species[27], although the sample size was limited. More recently, an emergency vaccination programme was launched for more than 200 captive and free-living California condors (*Gymnogyps californianus*), after HPAI-related die-offs had been detected in a wild population. In that case, the vaccine was a H5N1 subtype, reverse genetics-derived, inactivated product approved for chickens in the USA, and heterologous to the currently circulating 2.3.4.4b clade viruses[28]. Immune response, vaccine safety and practicability were first tested on the black vulture (*Coragyps atratus*) as a surrogate species and on captive individuals[10]. This latter study described an antibody persistence up to 42 days and suggested the need for a second boost-injection upon recapture[10].

The use of vaccination as a conservation strategy for wild populations in natural settings raises ethical, practical and technical challenges. In addition to the availability of safe vaccines leading to an appropriate immunological response, practical issues related to the ecology and biology of the targeted host species have been identified as major challenges[29]. It may then be especially important to identify key model species that can be studied in natural conditions and considered as surrogates for related endangered species facing conservation issues. Conducting experiments in natural settings provides the opportunity to practically assess the possible implementation of vaccination protocols. In some situations, carefully designed vaccination trials could enable the monitoring of wild individuals over periods of months, while ensuring suitable animal welfare conditions (limited handling, no artificial housing, natural feeding…). It could also allow to quantify the immune response of individuals in their natural environment, including difficult-to-control extrinsic factors, such as natural feeding, harsh winter weather conditions, exposure to predation risks and other infectious agents, that are usually not considered in vaccination trials using captive animals. Monitoring the serological status of individuals, and the local epidemiological situation in the field, would also enable to assess the practical feasibility of post-vaccination surveillance in the event of anti-HPAI vaccination being used for conservation purposes[18].

The king penguin (*Aptenodytes patagonicus*) is a widespread and abundant species in the sub-Antarctic area, with an ecology potentially prone to high circulation of the HPAI virus given the size and density of its colonies, that can reach tens to hundreds of thousands individuals[30–32]. As a long-lived species, the slow pace of life of king penguins may be associated with long lasting immune processes, as predicted for some of those species[33,34]. In this context, the long chick-rearing period of the king penguin on land[30,35] (c.300 days) make the species especially suited for investigating the dynamics of the immune response of chicks to vaccination. This least concerned species according to IUCN[36] could thus represent a suitable surrogate for endangered penguin species, such as the Northern rockhopper

penguin (*Eudyptes moseleyi*), but also for related species sharing comparable life history traits and potential similar immune system particularities (*Diomedeidae*, *Procellariidae*…). We thus decided to investigate the immune response of king penguin chicks to vaccination against a H5Nx HPAI virus in a natural setting.

Planning a HPAI vaccination program on wild birds should be preceded by an evidence-based risk assessment and an integrated cost-benefit analysis adapted to each situation[18], including critical considerations of ecological and epidemiological issues, and the identification of a potentially suitable vaccine[29,37]. The choice of the tested vaccine was made in light of the recent developments in RNA vaccine technologies and their wide applications to humans and domestic animals. In particular, messenger RNA vaccines have been approved and massively used as a fast and efficient response to SARS-CoV-2 pandemic. Next generations of self-amplifying messenger RNA (saRNA, also known as "replicons") vaccine show a great deal of potential advantages. The principle is based on introducing the encoding of a gene (i.e., RNA), associated with the essential molecular elements required for its self-amplification: the host cell can thus produce the targeted protein itself[38]. Therefore, a minimal dose of saRNA vaccine could induce an efficient humoral and cellular response, leading to complete immunisation. In the case of fast evolving viruses, like Influenza viruses, saRNA vaccine could be efficiently used to match a recently circulating viral genotype. With regard to Influenza, this approach had been identified before the SARS-CoV-2 pandemic as a possible game changer[39]. In recent applications, saRNA vaccination was shown to provide an efficient protection against H5Nx virus in ducks[40] and domestic geese[41], and has subsequently been used to vaccinate millions of ducks in France since 2024[42]. The saRNA vaccine 'Respons AI H5' (Ceva Santé Animale) had notably been shown to reduce virus excretion and transmission, as measured under experimental settings in ducks[43]. This study had also shown that a boost-injection applied 28 days after the primo-injection led to detectable anti-H5 antibodiy levels in ducks, however decreasing relatively fast afterwards. Because they specifically target the H5 haemagglutinin protein, such vaccines enable the Differentiation of Infected from Vaccinated Animals (DIVA vaccines): individuals exposed naturally to an unspecific influenza A virus would produce antibodies against the virus nucleoprotein (NP), while only antibodies against the H5 haemagglutinin protein would be expected in vaccinated ones[44]. This DIVA property is particularly important for vaccination campaigns in wild settings as it would allow the monitoring of the natural field exposure to avian influenza viruses of vaccinated individuals[18]. Nevertheless, a critical constraint posed by RNA vaccines is that they require ultra-low temperatures (−80 °C) for long-term storage, which can add to other practical constraints, such as the need for a booster injection for instance.

In this study, we report the results of a vaccination trial using a saRNA vaccine against H5 HPAI clade 2.3.4.4b protein conducted on king penguin chicks in a sub-Antarctic colony of the Southern Indian Ocean. A group of chicks receives a primo-injection at about 45 days post-hatching (once emancipated from their parents), and a booster 37 days later, while a group of chicks is used as controls. Along 250 days of fine-scale monitoring, the vaccinated individuals show a high and persistent anti-H5 immune response, granting a strong seroneutralisation capacity against a H5N1 2.3.4.4b clade virus, up to fledging. We report no adverse effects due to the vaccination treatment on chick health and survival. Overall, our results suggest that the saRNA H5 vaccine could be a valuable and powerful tool for wildlife conservation efforts in the current context of panzootic HPAI.

## Results

### Safety of the vaccine
No short-term adverse effects on king penguin chicks have been observed following the intra-muscular primo-injection (t0) and the

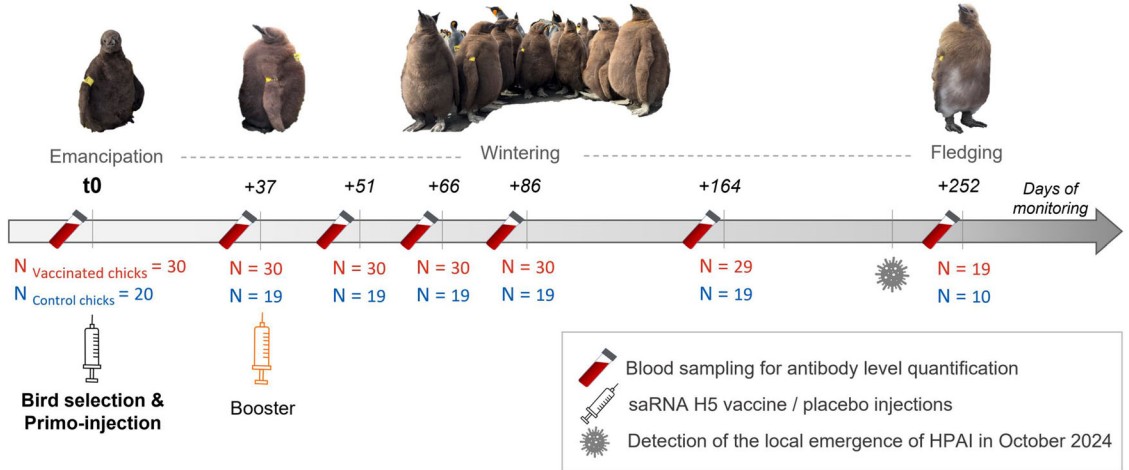

**Fig. 1 | Protocol of the vaccination trial conducted on king penguin chicks using a saRNA H5 vaccine.** The trial was conducted between February and November 2024 within the colony of Baie du Marin (Possession Island, Crozet archipelago), on emancipated chicks of about 45 days old (t0), monitored until fledging (t0 + 252 days). Sample sizes (N), reflecting the chick survival through the entire trial, refer to the number of vaccinated (red) and control (blue) chicks that were monitored. Pictures of chicks of different ages are displayed (Photograph credits: Camille De Pasquale and Romain Fischer, CC BY-NC-ND).

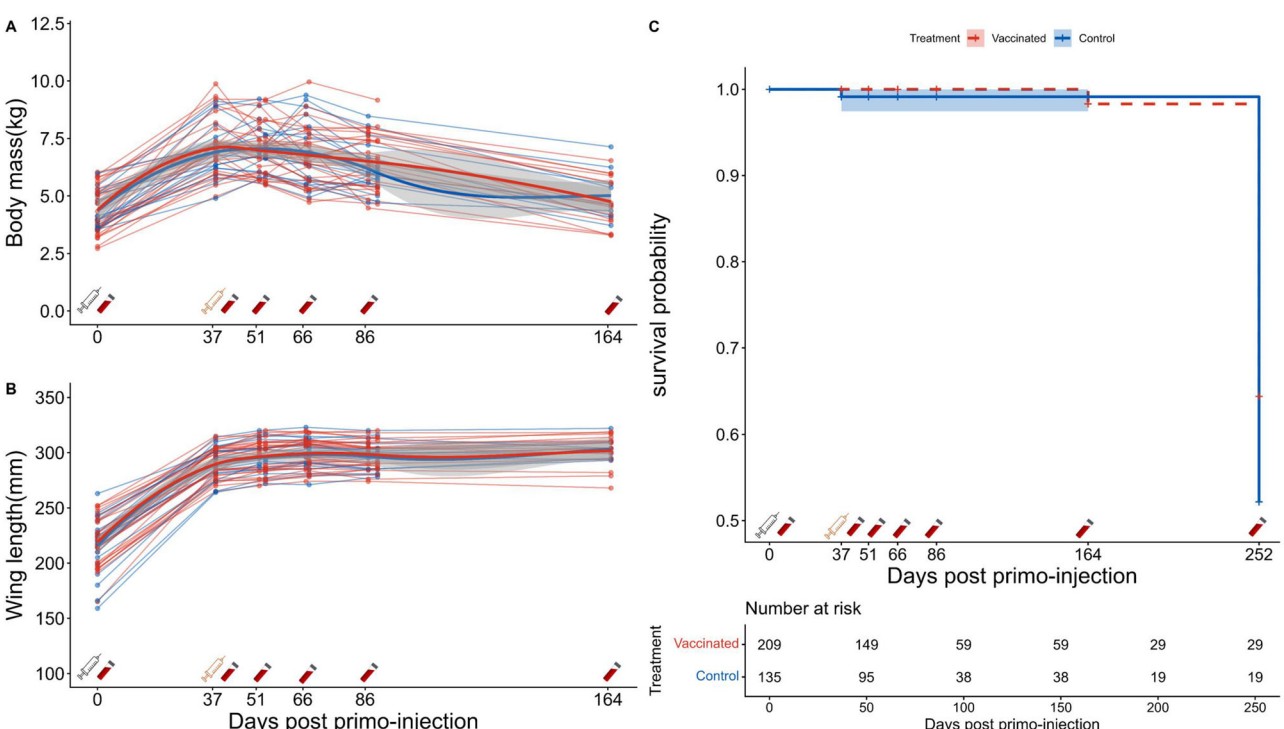

**Fig. 2 | Lack of adverse effects of saRNA H5 vaccination on king penguin chick survival and growth.** The changes of the body mass (**A**), flipper length (**B**) and survival (**C**) of the sampled chicks were recorded along the experiment and are represented using red for saRNA H5 vaccinated chicks, and blue for control chicks. Grey and yellow syringe symbols refer to the primo- and the booster-injection respectively. For each panel, the lines indicate the mean predicted value of the non-linear least-squares regression models and the shaded areas their 95% confidence intervals. **C** Kaplan–Meier survival curves compare vaccinated and control groups over time. The number of individuals at risk for the survival analysis is represented below the plot. Survival was compared using a two-sided log-rank test (*p* value = 0.35). There were no detectable differences in chick growth and survival between the two treatment groups.

boost-injection (t0 + 37 days) of the saRNA H5 vaccine (see Fig. 1 for details of the experimental design). Chick body mass and flipper length increased similarly with age in both treatment groups (Fig. 2A and B). The posterior distribution of the interaction between days post-injection and treatment group on both traits fell entirely within the predefined Region of Practical Equivalence (100% inside ROPE). This provides strong evidence for the absence of a biologically meaningful vaccination effect on chick body mass and flipper growth over time.

Chick survival rates remained very high for both treatment groups until t0 + 164 days (97% and 95% respectively for vaccinated and control chicks). As expected, some mortality occurred over the harsh Austral winter, but 19/30 vaccinated chicks and 10/20 control chicks could be recaptured for sampling at t0 + 252 days. Given the high expected recapture rate, this showed a minimum chick survival of at least 50% for both treatment groups (63% and 50% for the vaccinated and control group, respectively), from the primo-injection (chicks of

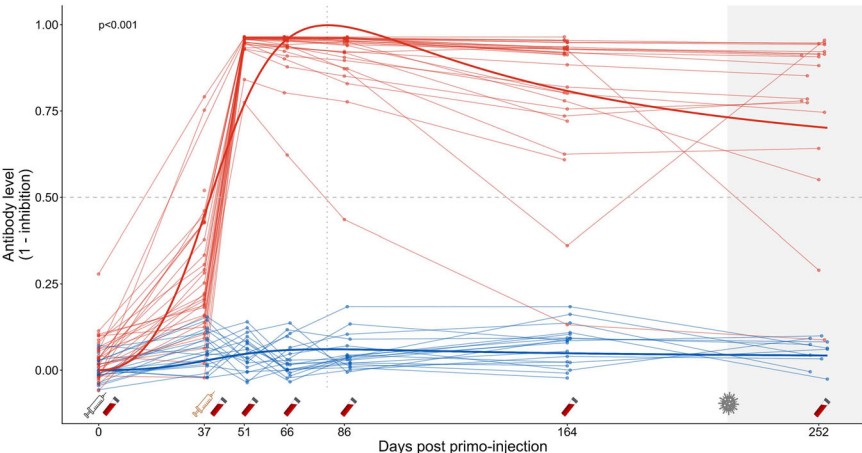

**Fig. 3 | Anti-H5 antibody responses over time of king penguin chicks from saRNA H5 vaccinated (red) and control groups (blue), measured with a H5 competitive ELISA assay.** Antibody levels (expressed as [1 – inhibition values]) are shown for individual chicks (points), plotted at their exact sampling day from primo-injection (day 0) until fledging (day 252). Thin lines represent individual trajectories, while thicker lines indicate group-level predictions from a non-linear least-squares regression model (nlsLM), with group-specific parameters following a Holling IV equation. The non-linear modelling analysis conducted to compare the dynamics of antibody responses between the vaccinated and control groups showed that the model that distinguished the two treatments had substantially better support (ΔAICc = 631). A positive value for the da parameter was highly significant ($t = 21.463$, $p < 0.0001$), showing a higher response in antibody level for the vaccinated group, and a higher antibody level over time following vaccination. Syringe symbols below the *x*-axis indicate vaccination events (grey: primo-injection at day 0; yellow: booster at day 37). Blood tube icons mark chick recaptures for blood sampling. The vertical dotted line represents the predicted timing of maximum antibody response obtained from the non-linear model. The horizontal dashed line shows the seropositivity threshold. The shaded grey area indicates the period where first HPAI cases were detected on Possession Island (from October 2024). The saRNA H5 vaccination induced a significant and persistent antibody response in king penguin chicks ($p < 0.001$).

c.45 days post-hatching) until just before fledging. Accordingly, no effect of the vaccination treatment on chick survival was detected over the period (Hazard Ratio HR = 1.50 [CI = 0.64-3.53, $p = 0.35$]) (Fig. 2C; and Supplementary Table 1).

## Anti-H5 antibody response following vaccination

All chicks from the vaccinated group showed a clear humoral immune response against the H5 protein, measured via a competitive H5 ELISA assay, while no anti-H5 antibodies were detected in control chicks along the whole trial (Fig. 3; and Supplementary Table 2; Supplementary Table 3; Supplementary Table 4). In order to account for a potential strong increase in anti-H5 antibody level at the start of the experiment, followed by a slow decay over time after reaching a peak, the dynamics of antibody levels of vaccinated and control birds were fitted using a Holling IV curve (Fig. 3), with the following equation (see Supplementary Table 5):

$$f(days) = \frac{(a_{control} + da \cdot treatment) \cdot days^2}{(b_{control} + db \cdot treatment) + (c_{control} + dc \cdot treatment) \cdot days + days^2}$$

(1)

The non-linear modelling analysis conducted to compare the dynamics of antibody responses between the vaccinated and control groups showed that the model that distinguished the two treatments had substantially better support (AICc$_{group}$ = −355.65, AICc$_{pooled}$ = 274.72, ΔAICc = 631). For the model distinguishing the two groups, a lower AICc was estimated (ΔAICc = 2.05) when the curve parameters db and dc were fixed to 0, indicating no difference between treatment groups in the offset and linear terms of the curve. A positive da value was highly significant ($t = 21.463$, $p < 0.0001$), showing a higher response in antibody level for the vaccinated group (~16x higher scaling response, a$_{control}$ ~ 0.03; a$_{vaccinated}$ = a$_{control}$ + da ~ 0.52), and a higher antibody level over time following vaccination. All parameters contributed meaningfully to the selected model (all *p*-values < 0.001). After the boost-injection, a higher anti-H5 antibody

level was measured in vaccinated chicks at t0 + 51 days, compared to levels measured at t0 + 37 days. The Holling IV curve indicated a predicted maximum antibody level reached at t0 + 83 days (−2*b$_{control}$/ c$_{control}$). Antibody levels of vaccinated chicks remained relatively high over time following the boost-injection, with 17/19 vaccinated chicks still seropositive at t0 + 252 days. The antibody levels of only two vaccinated birds decreased post boost-injection: one showed a rapid and permanent decrease, while the other showed a single intermediate value before increasing again at t0 + 252 days.

The analyses conducted on values resulting from the indirect H5 ELISA assay showed similar results (Supplementary Fig. 1; and Supplementary Table 2; Supplementary Table 6; Supplementary Table 7). The significant positive da value showed a higher response in anti-H5 antibody level (~16x higher scaling response, a$_{control}$ ~ 0.12; a$_{vaccinated}$ = a$_{control}$ + da ~ 1.99) for the vaccinated group, and a higher titre over time following vaccination (Supplementary Table 8). In addition, the fitted Holling IV curve appeared to predict a similar maximum antibody level reached at 83 days post primo-injection.

## Seroneutralisation response following vaccination

The linear mixed-effects model on chick plasma seroneutralisation level revealed a significant interaction between treatment and time ($\chi^2$ test, $p = 0.012$). Estimated marginal means showed that vaccinated individuals had substantially higher seroneutralisation titres than controls, at both 51 days (log(titre) = 11.13 ± 0.32 vs. 2.40 ± 0.46; $p < 0.0001$), and 252 days post primo-injection (log(titre) = 9.43 ± 0.32 vs. 2.60 ± 0.46; $p < 0.0001$) (Fig. 4; and Supplementary Table 9; Supplementary Table 10). Within-group comparisons revealed no significant change in seroneutralisation activity of control individuals between 51 and 252 days ($p = 0.98$), whereas vaccinated individuals showed a significant but moderate decline in antibody titres from 51 to 252 days (log(titre) = 11.13 ± 0.32 vs. 9.43 ± 0.32; $p = 0.0029$). These results show that the saRNA H5 vaccination provided an important seroneutralisation capacity to king penguin chicks, likely conferring an efficient level of protection of vaccinated chicks against HPAI virus along a large part of their rearing-period.

## Differentiating vaccinated individuals from potentially naturally exposed ones

The results of the ELISA analysis for anti-NP antibody showed that the levels remained relatively low and non-detectable for both treatment groups across the experiment, showing the lack of immune responses of the chicks to natural exposure to avian influenza A viruses (Fig. 5; Supplementary Table 2; Supplementary Table 11). It is important to note that the natural circulation of the HPAI H5N1 virus was first detected on the island on October 17th 2024, and more specifically on October 20th in the study colony[14], a few weeks before the last sampling point (t0 + 252 days). Abnormal die-offs of southern elephant seals (*Mirounga leonina*) associated to the detection of HPAI H5N1 virus in brain samples were observed, but also of adult king penguins[14]. The serological analyses nevertheless suggested that no chick had mounted a response until that last sampling point, possibly meaning that none had been exposed over the whole experiment trial.

## Discussion

Using vaccination for wildlife conservation requires to address a series of practical issues, a critical one being the identification of a safe vaccine that elicits a rapid and persistent immune response. This is the results we obtained using a saRNA vaccine developed against a H5 HPAI protein of clade 2.3.4.4b on king penguin chicks in a natural setting, over 250 days of field monitoring.

The saRNA H5 vaccine appeared to be safe, as no adverse effects were detected on chick health, survival and growth following injections of the vaccine. This is in accordance with previous experimental studies conducted with the same vaccine on ducks in captivity[40,43]. We report here a persistent antibody level against the H5 protein in the king penguin, in accordance with a recent study investigating antibody persistence in several wild species vaccinated with a HPAI replicon vaccine in captive settings[45]. Our results thus confirm the possibility of obtaining long antibody persistence in penguins against H5 HPAI[27]. They also corroborate the need for a second-injection to induce a higher immune response, as reported by previous studies[10,43], although we did not explore the long-term effect of a single injection. The high plasma seroneutralisation capacity of vaccinated chicks highlights a potentially strong and persistent protective effect against HP H5N1 virus along the chick-rearing period.

Although the experiment did not include a virus challenge, the same saRNA H5 vaccine was shown to confer protection against experimental exposure to HPAI in captive ducks[40] and geese[41], and appeared to reduce viral shedding. Given the results we obtained on anti-H5 antibody and seroneutralisation levels, we could expect the vaccine to be protective and limit viral shedding in king penguins. Re-

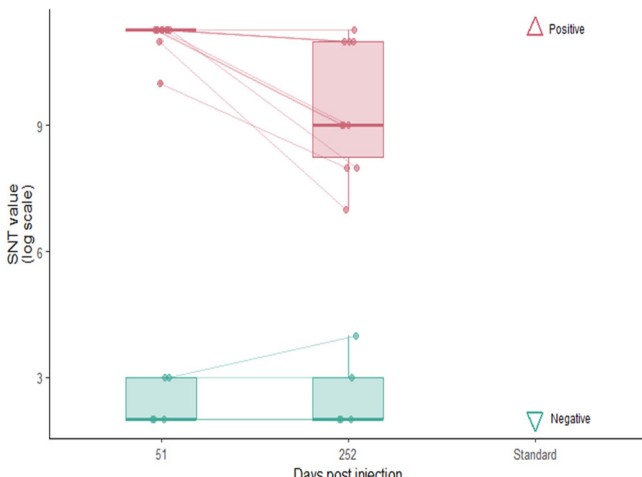

**Fig. 4 | Plasma seroneutralisation ability of saRNA H5 vaccinated and control king penguin chicks challenged with a HP H5N1 clade 2.3.4.4b virus.** Seroneutralisation titres (SNT, expressed as [log2(values)]) were measured on 10 vaccinated chicks (red points) and 5 control chicks (blue points) at two time points. Thin lines link the seroneutralisation capacity of the same bird, measured at both 51 and 252 days post primo-injection. All data points are shown, with horizontal jitter. The box plots indicate the 25th (Q1), median (50th), 75th (Q3) percentiles and inter-quartile Range (IQR = Q3–Q1), with whiskers extending to the most extreme data point that is no more than 1.5 × IQR from the edge of the box. Triangle symbols indicate values measured on plasma standards used as references to discriminate a high (positive standard, $n = 1$) and a low level (negative standard, $n = 1$) of seroneutralisation activity. The saRNA H5 vaccination induced an efficient immune response able to seroneutralise a H5N1 HPAI 2.3.4.4b virus ($p < 0.0001$), likely providing a persistent protection to king penguin chicks until fledging.

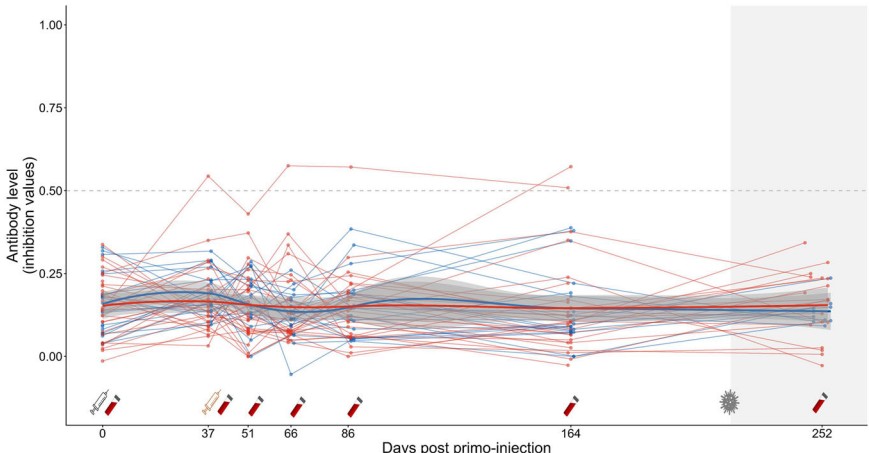

**Fig. 5 | Anti-NP antibody levels of king penguin chicks following either a saRNA H5 vaccination protocol (red dots, vaccinated chicks), or placebo injections (blue dots, control chicks).** Antibody levels (expressed as [1 − inhibition values]) are shown for individual chicks (points), plotted at their exact sampling day from primo-injection (day 0) until fledging (day 252). Thin lines represent individual trajectories, while thicker lines were generated using stat_smooth (method = *loess*) to show group-level trends, with shaded areas indicating 95% confidence intervals. The grey syringe symbol indicates the primo-injection (day 0), while the yellow one refers to the boost-injection, 37 days after. Blood tube icons mark chick recaptures for blood sampling. The horizontal dashed line indicates the seropositivity threshold. The shaded grey area indicates the period after the detection of the local emergence of HPAI in October 2024 on Possession Island. The plot shows a lack of detectable anti-NP antibodies for both treatment groups, indicating that sampled chicks had not mounted a response against any influenza viruses, suggesting that none had been infected by an influenza virus, even when HPAI H5N1 had started to be detected on the island.

running a comparable experiment at the same site now that active local circulation of the virus has been shown on Crozet[14] could enable exploring the efficiency of the vaccine by monitoring potential effects on viral shedding and protection against sub-lethal (e.g., growth) and lethal (i.e., survival) consequences of virus infection. However, this would only be correlative, as the level and timing of exposure to the virus would not be controlled. Alternatively, experimental viral challenge might be considered in confined settings in other surrogate species, notably to ascertain the correlation between measures of immune responses and protection.

A limitation of vaccination trials conducted in natural settings is the potential exposure of individuals to naturally circulating infectious agents, which could trigger concomitant immune responses. In the case of our study, we had checked that king penguins of the local population were unlikely to have been exposed to avian influenza viruses before the 2024 outbreak. A historical study had not detected antibody against avian influenza viruses on the colony in the nineties[46], and we did not find evidence of seropositive individuals against the NP-protein in king penguin adults sampled in the "Baie du Marin" colony in the 3 years prior the experiment (Supplementary Fig. 2), suggesting an absence of a natural circulation of influenza viruses in the colony. Similarly, all sampled chicks were seronegative at the start of the study, limiting potential issues with maternal antibody transfer and persistence[33]. Indeed, maternal antibodies transferred from mother to chick are known in poultry to confer protection to HP AIV and can interfere with some vaccine protocols[47]. A long persistence of maternal antibody level has been shown in a Procellariiform species[33,48], suggesting that temporal persistence of antibody level could vary among species. The transfer and temporal persistence of maternal antibody against H5 HPAI would be important to explore, given our results, as they could have implications for the protection of young chicks if females could be vaccinated prior to laying. Maternal antibodies that are transferred through the egg yolk in birds are IgY, not IgM[49], which might affect their ability to seroneutralise the virus. The results also stress the importance of exploring inter-year temporal persistence of anti-HPAI immune responses in adult seabirds[33].

The fact that the vaccine allowed a DIVA protocol is important because it means field surveillance could be efficient to inform managers about the exposure of individuals to AIV in relation to their history of vaccination. Despite the detected emergence of HP H5N1 clade 2.3.4.4b virus in the Baie du Marin colony in late October 2024[14], we did not detect any immune response of the studied chicks to avian influenza viruses over the period, which precluded us from inferring a potential protection conferred by the vaccine in naturally exposed individuals. The lack of detection of anti-NP antibodies in any chicks suggests that the virus had not widely spread among individuals within the colony by the time of the last sampling session (November 2024), and/or that recently exposed chicks would not have had time to mount a detectable immune response. Many wild species are susceptible to the virus and further monitoring of the circulation of HP AIV in wild animal communities, notably in communities of marine birds and mammals in subantarctic regions[14], would be especially important in this context. More generally, integrated approaches are required to face the threats posed by the HPAI H5N1 panzootic to the health wild species, domestic animals and humans[50], and vaccination represents a key possible mitigation tool.

A perceived limitation of conducting vaccination trials in natural settings is the difficulty of measuring the persistence of the immune response over a long time period, in contrast to studies conducted on captive animals[25,26]. The current study nevertheless allowed a robust experimental trial with the monitoring of naturally reared chicks over 250 days post primo-injection, which is larger than most studies in captive settings conducted on domesticated species[40]. We acknowledge though that the study design of this study was not appropriate for investigating longer temporal (inter-year) persistence of antibody

levels, which would be especially important for vaccine use for conservation. The long-term response of free-living individual seabirds to vaccines has been explored in a few species (e.g., vaccination against the Newcastle virus in Cory's shearwaters[48], and against avian cholera in yellow-nosed albatrosses[51]), but not in the case of an HPAI vaccine. Tracking king penguin chicks beyond 250-300 days post-hatching would be very challenging, as they leave the colony to fledge to sea. Moreover, the long-term use of external tags, such as flipper bands once the chicks have fledged, was shown to alter annual adult survival in this species[52]. In that case, using subcutaneous transponders and antennas to record returning individuals year after year could enable a long-term monitoring[52], although this is more efficient for adult individuals. King penguin colonies are very large and despite a relatively high degree of philopatry, it is indeed difficult to imagine recapturing fledged penguin chicks from the experiment to unable meaningful inference about antibody dynamics beyond the chick rearing period. Conversely, working on breeding adults in a specific part of the colony, where antennas are set up, could allow such investigations, as some individuals could be recaptured in different years. Conducting this type of experiment could be easier on flying seabird species as they can be ringed and resampled on their nest, as most species show a high level of philopatry[53,54]. Given the particularly threatened status of some populations of albatrosses[33], their longevity, site fidelity and likely susceptibility to the virus[13], conducting such an experiment in a large albatross species could be a next logical step.

The repeated sampling of the chicks over their whole rearing period after their vaccination provided the opportunity for a robust estimation of the dynamics of their antibody level. The observed variability in the dynamics of antibody levels in the vaccinated group suggests that some underlying factors, possibly genetic and/or environmental, may have affected the responses of a few individuals. This underlines the potential interest of considering larger sample sizes in future studies. The proportion of individuals that could be sampled repeatedly along a year could be strongly affected by external factors, such as predation risks, change in food availability and winter weather conditions. In the current study, the survival of the chicks was high but a proportion could not be recaptured after the winter due to unknown reasons. The last session of recapture of the chicks for sampling was carried out after the local emergence of HPAI on the colony[14], which may have lowered the probability of capturing individuals that may have still been alive due to restricted access to the colony for biosecurity reasons. The DIVA results nevertheless do not suggest that direct viral exposure of chicks occurred.

Considering vaccination against H5Nx HPAI as a mitigation strategy for the conservation of threatened species, the choice of the targeted population(s) should be based on the consideration of their endangered status, practical issues for vaccine delivery and the perceived threat from HPAI. This would require a solid understanding of the eco-epidemiological drivers of the risks of exposure of individuals and populations, and of their susceptibility to the virus. Determining the expected possibility of outbreaks in dense colonies and the possible exposure of individuals of high demographic value would notably be critical, as well as the role of the spatial structure of host populations on transmission risks[55]. The potential roles of reservoir and spreader hosts could also be important to consider. Some species, such as scavengers, may be particularly involved in the spatial spread of a disease, at local or larger scales, and their vaccination could also be considered to alter transmission. Avian scavengers and predators, like skuas, have for instance been suggested to be a prime contributor to the circulation of avian cholera agent within a seabird community[56], while giant petrels have been suggested to be involved in the long distance spread of infectious agents[57]. If vaccination had to be deployed, its potential effect on the evolution of the virus would also have to be considered[29]. This would depend on the vaccination coverage, the vaccine versatility, and the relative importance of host

populations for the maintenance of the virus. Further, this may dramatically depend on ecological characteristics of the host populations (size, range of movement, trophic behaviour, immune system), and on virus genotypes. Little is yet known about the transmission drivers of the HPAI virus among and within seabird and marine mammal host populations[5,7,14], but the integrated implementation of a combination of approaches (phylogenetics, serological monitoring, demography, environmental sampling at multiple sites and at different times over the course of the current epizootics, epidemiological modelling) has the potential to provide useful knowledge for informed risk assessment and mitigation measures. As for other species threatened by infectious agents, such as island scrub jay by the West Nile virus[58], or the Ethiopian wolf by rabies and canine distemper viruses[59], focused vaccination programs could then be considered.

## Methods

### Ethical issues and permits

Permits to implement the experiment were granted from the French Ministry of Research (APAFIS #45548-2023102712468223) after evaluation by the Comité Régional d'Ethique d'Occitanie and from Préfecture des TAAF (A-2024-95) after evaluation by the Conseil de l'Environnement Polaire and Conseil National de la Protection de la Nature. After the suspicion of HPAI emergence on Crozet, permits were granted by the Préfecture des TAAF to pursue eco-epidemiological fieldwork (A-2024-144).

### Model species and study population

The king penguin is a long-lived colonial seabird, breeding in colonies that can reach tens to hundreds of thousands of individuals, with an extended reproductive cycle of ca .14 months[35,60]. A single egg is laid, followed by an incubation time of 54 days and an extended rearing period of c.11 months. Partners share alternatively the incubation of the egg, then the care of the chick, until its full thermal emancipation (around 30–40 days post-hatching). Chicks are then left ashore in the colony by their parents for several months before fledging to sea at around 300 days post-hatching. Once the chicks are emancipated, they often gather in groups ('crèches') to withstand the harsh conditions of the Austral winter. Over that time period, they can be captured, resighted and recaptured, allowing good conditions for their monitoring over several months[61].

The experimental vaccination trial on king penguin chicks was conducted from February to November 2024 on Possession Island (Crozet archipelago, 46° 24′ 41″ S, 51° 45′ 22″ E, Southern Indian Ocean), within the colony of Baie du Marin.

### saRNA vaccine and practical issues

The saRNA RESPONS AI H5 vaccine (Ceva Santé Animale) was used to explore the immune response of the king penguin chick against a H5 HPAI clade 2.3.4.4b protein in natural settings. The vaccine was provided by the producer under a Material Transfer Agreement (MTA) for exclusive use in the trial. Stock vaccine vials were transported and stored at −80 °C during the whole travel to reach the lab facilities on Possession Island, until processing for the field experiment.

Prior to any injection, stock vaccine vials were placed at 4 °C for 12 h before each session of injections, to ensure a complete defrosting. The thawed stock vaccine solution was then diluted freshly with a provided diluent according to the manufacturer's guidelines. The diluted vaccine was administered within the recommended 2 h post-dilution (0.2 ml intra-muscular injection of 3 μg in the thigh). The use of the saRNA vaccine under field conditions in sub-Antarctica was facilitated by the proximity of the study colony to the scientific station, where −80 °C storage facilities were available. However, the possibility to keep the stock vaccine vials at 4 °C during 24 h before conducting a session of injections could allow some flexibility to organise vaccination trials in remote colonies.

### Experimental vaccination design

For the experiment, 50 emancipated chicks of approximately 45 days post-hatching were selected and randomly assigned to two treatment groups, and they were monitored along their rearing period, until fledging to sea. Those chicks were temporally captured and tagged in batches using alphanumeric plastic flipper bands and coloured flags allowing efficient and individual tracking during the whole chick rearing-period. Microtransponders (BioTherm13® 13 mm 134.2 kHz ISO FDX-B, Biomark, Idaho, USA) were subcutaneously inserted into the back of the neck, each one related to a unique and permanent identification number. Birds received either a primo-injection (t0) of the saRNA H5 vaccine in the thigh muscle (n = 30 chicks, vaccinated group), or a placebo injection using the vaccine diluent (n = 20 chicks, control group). A boost-injection was applied approximately 37 days later to all vaccinated chicks, while control chicks received a second placebo injection, under the same handling conditions as at t0. The bird immune responses were monitored along the entire rearing period through blood sampling at 7 capture times. For clarity of presentation and statistical analyses, sampling days were grouped into seven categories: t0, t0 + 37 days, t0 + 51 days, t0 + 66 days, t0 + 86 days, t0 + 164 days and t0 + 252 days (Fig. 1; and Supplementary Table 12). Blood samples (0.2 to 1 ml) were collected from the ulnar vein using a 1 ml heparinized syringe. Plasmas were separated from red blood cells by centrifugation within a few h. They were immediately frozen and stored at −20 °C before conducting serological analyses. All chick external tags (coloured flags and flipper bands) were removed at the end of the experiment, at t0 + 252 days. Both groups of control and vaccinated penguin chicks likely contained animals of both sexes, but immune responses of the animals were not differentiated by sex. The immune response of the mixed-sex animal group as a whole was analysed.

### Evaluation of the vaccine safety

To assess potential adverse effects of the vaccine, we first evaluated local short-term effects after each intra-muscular injection: chicks were kept a few minutes before release to check for any sign of limping, local swelling, or typical distress behaviour (hyperventilation, lack of reactivity…). Chick condition was also monitored at each capture. The chick survival rate in both treatment groups was evaluated regularly along the entire trial through remote checks from the edge of the colony, until all chick external tags were removed at t0 + 252 days. We assessed potential long-term adverse effects by monitoring the change in chick body mass and body size (flipper length), as morphometric indicators of chick growth, until t0 + 164 days. In the context of the HPAI outbreak detected locally in late October 2024 on Possession Island, we could not record the chick body masses/sizes at the last sampling point (t0 + 252 days) due to biosecurity restrictions. To evaluate whether vaccination affected chick body growth trajectories, we used a Bayesian linear mixed-effects model implemented in the brms package (version 1.22.0) of the R software[49]. The model included body mass or flipper length as the response variables, with days post-primo-injection, treatment group (vaccinated vs. control), and their interaction as fixed effects. Chick identity was included as a random intercept to account for repeated measures of individuals. To formally assess the absence of a biologically meaningful difference between treatment groups over time, we used a Region of Practical Equivalence (ROPE) approach. The ROPE for the interaction term (days: treatment) was set to ±0.2 kg. Posterior distributions and ROPE inclusion percentages were computed using the bayestestR package (version 0.16.0).

To evaluate whether the vaccination treatment had affected the chick survival over time, we fitted a Cox proportional hazards model with treatment group (vaccinated vs. control), as a categorical predictor. Time-to-event was measured from t0 to the last capture at t0 + 252 days, when external identification of individuals where removed. We used the coxph() function from the survival package of the R software (version 3.8-3). We assessed the proportional hazards

assumption using Schoenfeld residuals. Model results are reported as hazard ratio (HR) with 95% confidence interval (CI). A two-sided *p*-value < 0.05 was considered statistically significant.

## Quantification of specific antibody levels

The production and temporal persistence of specific H5 antibody levels were measured in plasma samples using commercially-available enzyme-linked immunosorbent assays (ELISA) kits. First, plasma samples were analysed using a competitive ELISA assay targeting the H5 protein of the clade 2.3.4.4b virus (H5-cELISA) and suitable for the monitoring of vaccination of domestic and wild birds (ID Screen® Influenza H5 Antibody Competition 3.0 Multi-species, reference no. FLUH5V3-5P, Innovative Diagnostics SARL, Grabels, France). Given previous results obtained on other seabird species suggesting that anti-chicken conjugate antibodies could bind with seabird antibodies[62], we complemented our results using an anti-H5 indirect ELISA assay (H5-iELISA) (ID Screen® Influenza H5 Indirect, reference no. FLUH5S-2P, Innovative Diagnostics SARL, Grabels, France). In order to evaluate the natural exposure of chicks to unspecific influenza A viruses potentially circulating in the colony (DIVA approach), we also tested plasma samples using a competitive ELISA assay targeting the non-specific nucleoprotein (NP-cELISA) (ID Screen® Influenza A Antibody Competition Multi-species, reference no. FLUACA-5P, Innovative Diagnostics SARL, Grabels, France). ELISA assays were conducted according to instructions provided by the manufacturer. Plasma anti-H5 and anti-NP antibody levels were obtained by reading absorbance values at 450 nm using a microplate reader (Tecan Infinite® 200 Pro; Tecan Group Ltd., Mannendorf, Switzerland), and correcting raw absorbance values using optical densities (OD) of positive (PC) and negative (NC) controls provided for each kit ($OD_{PC}$ and $OD_{NC}$, respectively). Anti-H5 and anti-NP antibody levels obtained with competitive ELISA are expressed as inhibition values ($inhibition_{sample} = OD_{sample}/OD_{NC}$), while anti-H5 antibody levels obtained with indirect ELISA are expressed as S/P ratio ($S/P_{sample} = OD_{sample}/OD_{PC}$). We referred to each threshold value provided by each ELISA kit instructions to discriminate seropositive from seronegative samples: H5-cELISA = 0.50; H5-iELISA = 0.50; NP-cELISA = 0.45. Analyses conducted on repeated sub-samples allowed us to assess the between- and within-assay repeatability for each ELISA test (see Supplementary Table 13 for detailed coefficients of variation).

## HP H5N1 seroneutralisation assay

A seroneutralisation assays was ran on a subset of plasma samples to explore the capacity of specific antibodies to seroneutralise a HPAI H5 virus of clade 2.3.4.4b. The assay was conducted under Biosafety Level 3 conditions using the A/chicken/France/D2107428/2021_2.1b H5N1 HP virus from 2.3.4.4b clade. Plasmas were heat-inactivated (56°C for 60 min) and twofold serial dilutions were performed in a 50-μl volume in microtiter plates. Sera obtained from experimentally immunized chicken with the homologous inactivated (Sp_H5N1_A/chicken/France/D2107428/2021_1) (positive control serum) and obtained from SPF chicken (negative control serum) were also used. Diluted plasmas were mixed with an equal volume containing 100 TCID50 of the H5N1 virus. Four control wells of virus alone (positive for the cytopathogenic effect (CPE) control) and four control wells of dilution medium alone (negative CPE control) were included. After 1h incubation at 37°C and 5% CO2 for neutralisation, 100 μl of MDCK cells at 2 × 105/ml were added to each well. Plates were incubated at 37°C and 5% CO2 for 72h, after which the CPE was read under the microscope. For each plasma, the seroneutralisation titer was expressed as the mean of the last dilution factor for which no CPE was observed of two independent experiments. The positive control serum has a titer greater than 2048, the negative control serum has a titer less than 8.

## Antibody levels comparison

Statistical analyses were conducted using R[63] (R Core Team, 2023, version 4.3.1). We compared the antibody level at each time point between vaccinated and control individuals using analyses of variance. The same model was used for both competition and indirect ELISA results. To assess the effect of vaccination on antibody dynamics, we used linear mixed-effects models (LMMs) fitted with the lme4 package in R. Antibody levels were modeled as the response variable, with treatment group (vaccinated vs. control), sampling day (as a factor), and their interaction as fixed effects. Individual identity was included as a random effect to account for repeated measures.

We tested the significance of main effects and interactions using Type III ANOVA with Satterthwaite's approximation for denominator degrees of freedom, implemented in the lmerTest package. When significant interactions were detected, we conducted post-hoc pairwise comparisons between treatment groups at each sampling day using estimated marginal means (EMMs, emmeans package), with Tukey adjustment for multiple testing.

## Fitted models for antibody levels

To model the dynamics of anti-H5 antibody levels, we fitted the sequential values from each individual with a Holling IV rational function[64] in order to account for the expected sigmoid early build-up of antibody level following the primo- and boost-injections, followed by a decay of antibody level towards a potential asymptotic value for large numbers of days post vaccination. A fixed treatment effect was set to compare the vaccinated and control groups (stable low level was expected for control chicks).

We fitted a non-linear regression model to the time series data (exact time series) using the nlsLM() function from the minpack.lm R package (version 1.2-4). The Holling IV function adapted to our modelling situation is:

$$f(\text{days}) = \frac{a \cdot \text{days}^2}{b + c \cdot \text{days} + \text{days}^2} \tag{2}$$

with f(days), the antibody level (calculated from ELISA titre values); days, the number of days since the primo injection; a, the value of the function at the asymptotic reached at high values of days; (−2b/c), the value of day at the peak antibody level. We compared a pooled model, in which vaccinated and control groups shared the same temporal response function, to a group model in which curve parameters were allowed to vary between the two groups (vaccinated vs placebo).

We compared three competing models:

- Pooled model: vaccinated and control chicks shared the same response curve (no group effect);
- Full group model: curve parameters were allowed to differ between groups, with the following parameterization:

$$f(days) = \frac{(a + da \cdot treatment) \cdot \text{days}^2}{(b + db \cdot treatment) + (c + dc \cdot treatment) \cdot days + \text{days}^2} \tag{3}$$

where days is the number of days since the primo-injection and treatment is a binary indicator of group membership (0 = control, 1 = vaccinated). We parameterized the Holling type IV function such that the control group served as the reference, with parameters $a_{control}$, $b_{control}$ and $c_{control}$. For the vaccinated group, parameters were expressed as additive deviations from the control: $a_{control}$+da, $b_{control}$+db, and $c_{control}$+dc.

- Reduced group model: a nested version of the group model where only the asymptote differed between groups (da ≠ 0, db = dc = 0).

Model comparisons were conducted using the corrected Akaike Information Criterion (AICc) to account for differences in model complexity and sample size, using the AICcmodavg R package (version 2.3-4). The model with the lowest AICc value was retained as the most

parsimonious representation of the data. A difference in AICc greater than 2 was considered evidence of meaningful support for the better-fitting model.

### Reporting summary

Further information on research design is available in the Nature Portfolio Reporting Summary linked to this article.

## Data availability

We provide in Supplementary Data 1 details of the vaccination experiment data regarding serological analyses at each sampling time for the two treatment groups (control/vaccinated), in Supplementary Data 2 the data on chick morphometrics, and in Supplementary Data 3 data of the seroneutralisation assay.

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

## Acknowledgements
We acknowledge critical support from the French Polar Institute (IPEV ECOPATH-1151 [TBo] and ECOENERGY-119 [AS]), ANR AAPG ECOPATHS and WILDFLU projects (ANR-21-CE35-0016 [TBo], ANR-25-CE35-0691 [TBo, BG]), and Ceva Wildlife Research Fund. The study was also facilitated by supports as part of REMOVE_DISEASE project (ANR-21-BIRE-0006 [TBo]; Biodiversa+ and Water JPI joint call for projects under the BiodivRestore ERA-NET Cofund GA N°101003777), ECOPOP OSU OREME, Zone Atelier Terres Australes et Antarctiques (ZATA) and CNRS Ecologie & Environnement SEE-Life ECOPATH and ECONERGY. We also acknowledge support from Direction de l'Environnement of TAAF. We conducted the ELISA serological analyses at the GEMEX (CEFE) platform. We thank Zohria-Lys Guillerm, Samuel Laporte, Norith Eckbo, Blaise Raymond, Guillaume Lespagnol, Julie Carvalho, Lucia Llorente, Marine Delmas, Jean-Sebastien Bugand Bugandet, Alan Allart, Martin Meyer, Romain Fischer, Aude Noiret and Bastien Bauger for help in the field and Vincent Viblanc, Pierre-Marie Borne, Gwenaelle Dauphin, Florent Lavigne and Alice Mistou for their help at various stages of the study.

## Author contributions
T.Bo., M.L., J.T., T.Br. and A.S. conceived the study and led the field work; C.D.P. and E.M. implemented the experiment; M.L. performed the serological analyses and BG and PM the seroneutralisation assay; J.T., M.L. and T.Br. conducted the statistical analyses; T.Bo. and M.L. wrote the first version of the manuscript and all authors provided inputs.

## Competing interests
The authors declare as only conflict of interest that the work was partially funded by Ceva Wildlife Research Fund, an endowment fund created by Ceva Santé Animale, whose objective is to support applied animal health research projects contributing to wildlife conservation. Ceva Wildlife Research Fund had no role in the conceptualization, design, data collection, analysis, decision to publish, or preparation of the manuscript.
