## [Transparent Peer Review file · Nature Communications]

Vaccination against H5 HP influenza virus leads to persistent immune response in wild king penguins

Corresponding Author: Dr Thierry Boulinier

Version 0:

Reviewer comments:

Reviewer #1

(Remarks to the Author)

The manuscript by Mathilde and co-authors investigates the immune response of king penguin chicks vaccinated with an H5Nx highly pathogenic avian influenza (HPAI) virus vaccine under natural environmental conditions. This is a study of considerable significance. The authors demonstrated both the safety of vaccination in king penguin chicks and the persistence of IgG and neutralizing antibody responses for up to 252 days. These findings are consistent with previous studies validating the efficacy of saRNA vaccines in other target species, underscoring the broad application potential of AIV saRNA vaccines. Furthermore, differential diagnostic analyses confirmed that the king penguin chicks were not naturally infected with AIV during the experimental period, highlighting the value of DIVA vaccines in distinguishing natural infection from vaccine-induced immunity.

The manuscript is very well written, particularly in its detailed description of the materials and methods. Overall, the findings are robust and provide valuable insights that may benefit future H5 vaccine research in other endangered penguin species. Some of concerns are outlined below.

1. As the authors noted, a key scientific limitation of this study is the absence of a challenge experiment. This omission poses a challenge in establishing the correlation between antibody titers and protective efficacy, which should be further evaluated under controlled laboratory conditions in future studies.
2. Considering that neither the authors nor previous studies have detected AIV infection in king penguins (DOI: 10.1007/s00300-009-0587-4; DOI: 10.1111/j.1751-0813.1981.tb05839.x), it remains uncertain whether this species is susceptible to HP H5N1. This question warrants further investigation and ongoing surveillance, as local outbreaks may pose potential threats to the species and could directly affect the future applicability and value of the vaccine. Some discussions need to be strengthened, such as the fact that certain wild animals can be infected with the H5 subtype of avian influenza, and the broad host range of the H5 subtype should be addressed and discussed with references from top-tier journals. Such as : doi: 10.1016/j.cell.2025.08.016 doi: 10.1016/j.cell.2024.07.024; doi: 10.1016/j.cell.2023.04.002
3. The initial sample size was relatively small and further decreased by the end of the experiment, which may have affected the assessment of antibody persistence. This concern is supported by Figure 3, in which one vaccinated individual exhibited antibody levels comparable to those of the control group.
4. Although the authors discussed the challenges of assessing long-term antibody persistence (e.g., over one year), it is important to consider that penguins are long-lived species. Therefore, the duration of antibody responses should ideally be evaluated through to adulthood or the breeding period. Alternatively, the title could be refined to specify the actual time frame covered in this study, for example: "...persistent immune response in wild king penguin fledglings prior to sea departure."
5. A minor issue in the figures is that the use of light red and light green makes it difficult to distinguish overlapping data points. It is recommended that the color of the vaccinated group be changed to a darker shade to improve visual clarity and differentiation.

Reviewer #2

(Remarks to the Author)

This manuscript describes the field evaluation of an H5N1 influenza saRNA vaccine in penguins. The impact of the work lies in the fact that the H5N1 virus is decimating wild bird populations, it is encroaching upon likely naïve bird populations in the Antarctica region, and interventions are being sought. Data supporting immunization of wildlife against H5N1 influenza is scant and this manuscript is therefore highly significant, important, and will be of broad interest.

Overall the data presented is simple and clear and the conclusions drawn are sound.

The following are minor comments for the authors' consideration.

1. Pg7, ln 186. It is a little unclear if survival rates presented are a result of bird deaths or bird deaths and/or inability to recapture specific animals. I realize this may be a difficult question for the authors to definitively answer, but some general comment on expected recapture rates in the absence of likely death would be useful. I guess the question is "how hard is it to recapture all birds in a study group this size".
2. Ln 244. 1 out of 19 birds did not appear to respond to boosting. At face value, 5% of non-responders is large. Was there anything in particular noted about the non-responding bird (low body weight, issues noted with vaccination etc) that might help explain this result?
3. What is missing from the manuscript is a way to link the observed antibody titers with expected protection. Is there any way to infer this link? Possibilities could be the use of a hemagglutination inhibition assay where there is an accepted protective correlate (in humans and chickens at least). Or correlation of the titers detected in this paper with protections observed in the captive duck and geese studies that were conducted with the same vaccine.
4. More detail is needed on the seroneutralization assay. For example, what infectious dose of virus was used, how were titers read etc.

Version 1:

Reviewer comments:

Reviewer #1

(Remarks to the Author)
I have no other comments.

Reviewer #2

(Remarks to the Author)
the authors have responded well to my suggestions.

POINT-BY-POINT RESPONSES TO THE REFEREES' COMMENTS: AUTHORS RESPONSES IN BLUE BELOW, with line numbers for the file with track changes.

REVIEWER COMMENTS

Reviewer #1 (Remarks to the Author):

The manuscript by Mathilde and co-authors investigates the immune response of king penguin chicks vaccinated with an H5Nx highly pathogenic avian influenza (HPAI) virus vaccine under natural environmental conditions. This is a study of considerable significance. The authors demonstrated both the safety of vaccination in king penguin chicks and the persistence of IgG and neutralizing antibody responses for up to 252 days. These findings are consistent with previous studies validating the efficacy of saRNA vaccines in other target species, underscoring the broad application potential of AIV saRNA vaccines. Furthermore, differential diagnostic analyses confirmed that the king penguin chicks were not naturally infected with AIV during the experimental period, highlighting the value of DIVA vaccines in distinguishing natural infection from vaccine-induced immunity.

The manuscript is very well written, particularly in its detailed description of the materials and methods. Overall, the findings are robust and provide valuable insights that may benefit future H5 vaccine research in other endangered penguin species.

AUTHOR RESPONSE: We thank the referee for these very positive comments.

Some of concerns are outlined below.

1. As the authors noted, a key scientific limitation of this study is the absence of a challenge experiment. This omission poses a challenge in establishing the correlation between antibody titers and protective efficacy, which should be further evaluated under controlled laboratory conditions in future studies.

AUTHOR RESPONSE: We agree that this is a scientific limitation of the study, but it is difficult to imagine to conduct a virus challenge in controlled conditions for ethical and practical reasons given the considered penguin species. We now outlined in the Discussion that the correlation between antibody titers and protective efficacy could need to be further evaluated under controlled laboratory conditions, most conveniently in another species, by inserting 'Alternatively, experimental viral challenge might be considered in confined settings in other surrogate species, notably to ascertain the correlation between measures of immune responses and protection' (lines 347-349).

2. Considering that neither the authors nor previous studies have detected AIV infection in king penguins (DOI: 10.1007/s00300-009-0587-4; DOI: 10.1111/j.1751-0813.1981.tb05839.x), it remains uncertain whether this species is susceptible to HP H5N1. This question warrants further investigation and ongoing surveillance, as local outbreaks may pose potential threats to the species and could directly affect the future applicability and value of the vaccine. Some discussions need to be strengthened, such as the fact that certain wild animals can be infected with the H5 subtype of avian influenza, and the broad host range of the H5 subtype should be addressed and discussed with references from top-tier journals. Such as : doi: 10.1016/j.cell.2025.08.016 doi: 10.1016/j.cell.2024.07.024; doi: 10.1016/j.cell.2023.04.002

AUTHOR RESPONSE: There is indeed no evidence that king penguins on Crozet had been exposed to avian influenza virus *before 2024* (see the two reference cited by the referee), but as we outlined in the manuscript the virus emerged locally on Crozet in October 2024 and was associated with die-offs of elephant seals and king penguins, as we reported in Clessin et al. 2025 (Circumpolar spread of

avian influenza H5N1 to southern Indian Ocean islands. Nature Communications 16: 8463. doi.org/10.1038/s41467-025-64297-y), which was available only as a pre-print at the time we had submitted the current Lejeune et al. manuscript. The detection of H5N1 HP avian influenza of clade 2.3.4.4b virus in the brain of several individuals, together with individuals showing symptoms suggestive of HP avian influenza infection, makes us confident that this species is susceptible to HP H5N1. We do agree with the referee that it is of outmost importance to continue investigation on this issue, notably to evaluate what proportion of individuals of this species have been exposed and mounted an immune response, and also how other wild species, notably in sub-Antarctic marine bird and mammal communities, may be exposed, and to infer between-species transmission pathways (we have much on-going work on those issue, with sampling efforts at broad spatial scales). We are now specifying in the last part of the Results that 'Abnormal die-offs of southern elephant seals (*Mirounga leonina*) associated to the detection of HPAI H5N1 virus in brain samples were observed, but also of adult king penguins[14]' (lines 310-312). Further, we now stressed in the Discussion that many wild species appear susceptible to HP H5Nx viruses, which warrant further extensive monitoring of the eco-epidemiological situation, with implications for the potential use of a vaccine as a mitigation tool (with a citation of Moratorio et al. 2024 *Cell*, as suggested by the referee; lines 381-387).

3. The initial sample size was relatively small and further decreased by the end of the experiment, which may have affected the assessment of antibody persistence. This concern is supported by Figure 3, in which one vaccinated individual exhibited antibody levels comparable to those of the control group.

AUTHOR RESPONSE: The sample size was constrained by animal experimentation considerations, but it was clearly sufficient to show antibody persistence in a high proportion of individuals. The repeated sampling of individuals over the course of the experiment allowed us making a robust investigation of the antibody dynamics. The occurrence of one of the vaccinated individuals showing an early increase in anti-H5 antibody level, followed by a fast decline to an antibody level comparable to those of the control group, raises the question of the underlying variability of humoral immune response in this species, suggesting the interest of more extensive immunology studies. Following this remark, we now outline in the Discussion that larger sample sizes could be useful for this reason in future studies (lines 416-430).

4. Although the authors discussed the challenges of assessing long-term antibody persistence (e.g., over one year), it is important to consider that penguins are long-lived species. Therefore, the duration of antibody responses should ideally be evaluated through to adulthood or the breeding period. Alternatively, the title could be refined to specify the actual time frame covered in this study, for example: 'persistent immune response in wild king penguin fledglings prior to sea departure';

AUTHOR RESPONSE: We fully agree with the referee that the fact that the studied species is long-lived is of outmost importance. Given that such long lived species have deferred maturity (breeding for the first time at age 2 or 3 for penguins, or when more than ten years old in some albatross species), studying the duration of antibody responses through to adulthood in such wild species would be especially challenging (even when individuals would come back to breed, it would be difficult to find them among the very large number of conspecifics in the breeding colonies, especially in penguins that cannot be marked with outside marks). Studying antibody persistence among years in adult is indeed the next logical step. This is a perspective specifically discussed in a whole paragraph of the Discussion (lines 388-412) and now we added 'Given the particularly threatened status of some populations of albatrosses [33], their longevity, site fidelity and likely

susceptibility to the virus [13], conducting such an experiment in a large albatross species could be a next logical step'. (lines 412-415). Regarding a potential change to the title, we think that the current title conveys the main content of the manuscript while respecting the requested number of words (15), but we would be ready to change it to something like 'Vaccination against H5 HP influenza virus leads to persistent immune response in wild king penguins chicks' or 'Vaccination against H5 HP influenza virus leads to persistent immune response in wild king penguin fledglings prior to sea departure' if found more appropriate.

5. A minor issue in the figures is that the use of light red and light green makes it difficult to distinguish overlapping data points. It is recommended that the color of the vaccinated group be changed to a darker shade to improve visual clarity and differentiation.

AUTHOR RESPONSE: This can be changed accordingly.

Reviewer #2 (Remarks to the Author):

This manuscript describes the field evaluation of an H5N1 influenza saRNA vaccine in penguins. The impact of the work lies in the fact that the H5N1 virus is decimating wild bird populations, it is encroaching upon likely native bird populations in the Antarctica region, and interventions are being sought. Data supporting immunization of wildlife against H5N1 influenza is scant and this manuscript is therefore highly significant, important, and will be of broad interest.

Overall the data presented is simple and clear and the conclusions drawn are sound.

AUTHOR RESPONSE: We thank the referee for these very positive comments.

The following are minor comments for the authors' consideration.

1. Pg7, Ln 186. It is a little unclear if survival rates presented are a result of bird deaths or bird deaths and/or inability to recapture specific animals. I realize this may be a difficult question for the authors to definitively answer, but some general comment on expected recapture rates in the absence of likely death would be useful. I guess the question is 'how hard is it to recapture all birds in a study group this size';

AUTHOR RESPONSE: The expected recapture rate in the absence of likely death is close to 1 given the field situation of the colony (close to the base), the type of marking used on the chicks while they are on land, the resighting efforts and the skills of the field workers. Formal capture-mark-recapture analyses on an independent but comparable data set gathered in a previous year confirmed that the detectability was close to 1 if the individuals were still alive. If an individual was not recaptured, it had most likely died given the observation pressure, but we cannot be sure of this so we keep the statement that it is a minimum observed survival rate, but we now specify 'Given the high expected recapture rate,....' (lines 186-187). Of note, the situation at the end of the experiment was particular because the starting outbreak of HPAI (that had started in October 2024) had constrained accesses to the colony. We now outline this in the Discussion (lines 421-430).

2. Ln 244. 1 out of 19 birds did not appear to respond to boosting. At face value, 5% of non-responders is large. Was there anything in particular noted about the non-responding bird (low body weight, issues noted with vaccination etc) that might help explain this result?

AUTHOR RESPONSE: Nothing particular was noted about the non-responding bird that might help explain this result. We agree with the referee that exploring potential causes of non-responses in

such a wild species would be of value. We now mention this in the Discussion (lines 416-421) as this would be important to consider for further work on the topic.

3. What is missing from the manuscript is a way to link the observed antibody titers with expected protection. Is there any way to infer this link? Possibilities could be the use of a hemagglutination inhibition assay where there is an accepted protective correlate (in humans and chickens at least). Or correlation of the titers detected in this paper with protections observed in the captive duck and geese studies that were conducted with the same vaccine.

AUTHOR RESPONSE: It is always difficult to infer the expected level of protection with observed antibody titers, especially in a (wild) species for which little is known. The observed antibody titers are nevertheless high compared to what has been reported for captive ducks for which infectious challenges were conducted after vaccination with the vaccine we used (Grasland et al. 2023a and 2023b). They are also high compared to the captive geese vaccination study we cite that considered several vaccines among which the one we used (Piesche et al. 2025). Also, we present both anti-H5 ELISA results as well as seroneutralisation results against a 2.3.4.4b clade HP AI virus. We prefer not to go beyond the statements we are already making in the main text (lines 280-286, 298-300, 337-341, .

4. More detail is needed on the seroneutralization assay. For example, what infectious dose of virus was used, how were titers read etc.

AUTHOR RESPONSE: More detail is now provided on the seroneutralization assay. The following paragraph was added in the Methods section (590-606):

“A virus seroneutralisation assay was conducted under Biosafety Level 3 conditions using the A/chicken/France/D2107428/2021_2.1b H5N1 HP virus from 2.3.4.4b clade. Plasmas were heat-inactivated (56°C for 60 min) and twofold serial dilutions were performed in a 50- μ l volume in microtiter plates. Sera obtained from experimentally immunized chicken with the homologous inactivated virus (Sp_H5N1_A/chicken/France/D2107428/2021_1) (positive control serum) and obtained from SPF chicken (negative control serum) were also used. Diluted plasmas were mixed with an equal volume containing 100 TCID₅₀ of the H5N1 virus. Four control wells of virus alone (positive for the cytopathogenic effect (CPE) control) and four control wells of dilution medium alone (negative CPE control) were included. After 1h incubation at 37°C and 5% CO₂ for neutralization, 100 μ l of MDCK cells at 2×10^5 /ml were added to each well. Plates were incubated at 37°C and 5% CO₂ for 72h, after which the CPE was read under the microscope. For each plasma, the seroneutralisation titer was expressed as the mean of the last dilution factor for which no CPE was observed of two independent experiments. The positive control serum has a titer greater than 2048, the negative control serum has a titer less than 8.”

** See Nature Portfolio's author and referees' website at www.nature.com/authors for information about policies, services and author benefits.

This email has been sent through the Springer Nature Tracking System NY-610A-NPG&MTS
Confidentiality Statement: This e-mail is confidential and subject to copyright. Any unauthorised use or disclosure of its contents is prohibited. If you have received this email in error please notify our Manuscript Tracking System Helpdesk team at <http://platformsupport.nature.com>. Details of the confidentiality and pre-publicity policy may be found here <http://platformsupport.nature.com>

<http://www.nature.com/authors/policies/confidentiality.html>

<http://www.nature.com/authors/policies/confidentiality.html>

[Privacy Policy](http://www.nature.com/info/privacy.html) | [Privacy Policy](http://www.nature.com/info/privacy.html)

POINT BY POINT RESPONSES TO THE REVIEWERS' COMMENTS

Reviewer #1 (Remarks to the Author):

I have no other comments.

AUTHOR RESPONSE: We thank reviewer #1 for the re-review of our manuscript.

Reviewer #2 (Remarks to the Author):

the authors have responded well to my suggestions

AUTHOR RESPONSE: We thank reviewer #2 for the re-review of our manuscript.